# Origami Paper-Based Electrochemical (Bio)Sensors: State of the Art and Perspective

**DOI:** 10.3390/bios11090328

**Published:** 2021-09-10

**Authors:** Noemi Colozza, Veronica Caratelli, Danila Moscone, Fabiana Arduini

**Affiliations:** 1Department of Chemical Science and Technologies, University of Rome “Tor Vergata”, Via della Ricerca Scientifica, 00133 Rome, Italy; noemi.colozza@uniroma2.it (N.C.); veronica.caratelli@uniroma2.it (V.C.); moscone@uniroma2.it (D.M.); 2SENSE4MED, Via Renato Rascel 30, 00128 Rome, Italy

**Keywords:** DNA, enzyme, antibody, cell, molecularly imprinted polymers

## Abstract

In the last 10 years, paper-based electrochemical biosensors have gathered attention from the scientific community for their unique advantages and sustainability vision. The use of papers in the design the electrochemical biosensors confers to these analytical tools several interesting features such as the management of the solution flow without external equipment, the fabrication of reagent-free devices exploiting the porosity of the paper to store the reagents, and the unprecedented capability to detect the target analyte in gas phase without any sampling system. Furthermore, cost-effective fabrication using printing technologies, including wax and screen-printing, combined with the use of this eco-friendly substrate and the possibility of reducing waste management after measuring by the incineration of the sensor, designate these type of sensors as eco-designed analytical tools. Additionally, the foldability feature of the paper has been recently exploited to design and fabricate 3D multifarious biosensors, which are able to detect different target analytes by using enzymes, antibodies, DNA, molecularly imprinted polymers, and cells as biocomponents. Interestingly, the 3D structure has recently boosted the self-powered paper-based biosensors, opening new frontiers in origami devices. This review aims to give an overview of the current state origami paper-based biosensors, pointing out how the foldability of the paper allows for the development of sensitive, selective, and easy-to-use smart and sustainable analytical devices.

## 1. Introduction

In 2015, all United Nations Member States adopted the 2030 Agenda for Sustainable Development to provide a blueprint for peace and prosperity for people and the planet, both now and in the future. Analytical chemistry could carry out many activities to achieve the different sustainable development goals (SDGs), starting by ensuring healthy lives and promoting well-being for all at all ages (SDG3), through sustainable management of water and sanitation for all (SDG6), and conserving and sustainably using the oceans, seas, and marine resources, for sustainable development (SDG14), since the detection of biomarkers, pollutants, food quality indicators, among others are needed to achieve SDGs in a sustainable way. Nevertheless, if detection is important, it is also relevant how it must be carried out. If detection is accomplished by generating other pollutants, the detection avoids some sustainable key elements. The 11th Principle of Green Chemistry boosts the in situ analysis [1], but this feature is not enough. Thus, Green Analytical Chemistry has recently opened the concept of White Analytical Chemistry which extends and complements the Green Analytical Chemistry vision for giving coherence and synergy of the analytical, ecological, and practical attributes [2].

In this context, electrochemical (bio)sensors match several features such as the avoiding of organic solvents, the reduction of reagent consumption, the capability to measure on-site, and the reduced sample treatment, to name a few. A giant step was achieved by paper-based devices which have opened a new route in the sensing field, being at the state of the art the most eco-designed sensors.

Indeed, besides being plastic free, paper sensors are able to:Reduce the use of energy consumption (for instance, using the capillarity for the microfluidics instead of an external pump) [3];Reduce chemicals (since the reaction can happen in the few μL solution layer within the cellulose network) [4];Reduce or avoid the sample treatment by exploiting the porosity of the paper [5];Allow for the paper network as reservoir for modifying the sensors with nanomaterials [6,7];Deliver reagent-free analytical tools by exploiting the porosity of the paper [8,9];Carry out easily the multiplex analyses [10];Increase the sensitivity by multistep of sample loading and waiting to become dry before exploiting the 3D network of the paper [11];Overcome the limitation of alumina or polyester-based sensors (by measuring the analytes in gas phase of surface, without any additional external sampling system) [12,13,14].

In the paper-based devices field, if the Whiteside group was the pioneer in the colorimetric analysis [15,16,17], in the electrochemical field this primacy can be attributed to the Henry group [18], who combined the previously reported outstanding features of the paper with unrevealed features of the electrochemical detection, this type of transduction being characterized by high sensitivity, the capability to work in colored samples, and a connection with miniaturized commercially available transducers.

In the last 10 years, the research activity in this sector has had a rapid grown, attested by several reviews [19,20,21,22,23,24,25,26,27,28]. In 2020, a number of reviews appeared demonstrating that this is a hot topic, and if the first reviews are more general, the last ones are more particular such as potentiometric paper-based devices [28], attesting to the higher numbers of the articles published in this field. Regarding the configurations, if the first configuration of the paper-based devices is mainly based on the horizontal flow, recently the foldability has boosted the vertical microfluidics, opening for a further route in this paper-based device field and leading to the origami paper-based devices. The capability of paper to be cut and easily folded has motivated scientists to design several configurations allowing for multifarious and polyhedric devices for smart detection of several analytes. Herein, we reported, to our knowledge, the first review focused on electrochemical origami-paper based biosensors furnishing an overview of how the foldability features can generate interesting configurations of enzymatic, immuno-, DNA, cell, and molecular imprinting polymers (MIPs) biosensors, enhancing the sensitivity, selectivity, and easiness to carry out measurements (Table 1).

Our research covers the publication produced in the last ten years, going back to the first origami-like paper-based biosensor that has been reported. The papers have been selected by choosing the keywords “origami”, “paper”, “electrochemical biosensor” on the google scholar web search.

## 2. Origami Paper-Based Electrochemical Enzymatic Biosensors

In the overall scenario of biosensors, the electrochemical enzymatic biosensor is the type of biosensor widely investigated and developed, taking into account that the most famous electrochemical biosensor is the one based on the glucose oxidase enzyme. In the case of glucose oxidase biosensor, the target analyte is glucose, namely the enzymatic substrate, and thus the electrochemical output is proportional to the amount of the target analyte.

Within the paper-based device for glucose detection, an easy configuration has been reported by Liang et al. [29], where the capability to move the pad near the classical printed electrochemical sensor was used to eliminate the electrochemical interferences in the case of glucose detection. In detail, before folding the enzymatic pad (thus when the enzyme is not present on the working electrode surface), the interferents such as ascorbate, urate, and paracetamol were completely consumed by a simple electrolysis step. Then, the enzymatic pad, with the enzyme loaded by drop casting, was put into contact with the working electrode allowing for the coulometric detection of glucose in the range of 0 to 24 mM, covering the diabetic range with recovery comprised in the range of 98–102%.

Then, the connection of the enzymatic pad to the working electrode allowed for the coulometric detection of glucose in the range from 0 to 24 mM, covering the diabetic range with recovery comprised in the range of 98–102%.

The configuration reported by Li et al. [30] combined two layers, encompassing one layer for the electrochemical cell, which was drawn with a pencil, and one layer for the enzymatic pad, containing glucose oxidase. In this case, the porosity of the paper was used to load both the glucose oxidase on the enzymatic pad and the electrochemical mediator, namely ferrocenecarboxylic acid, on the electrochemical cell layer. To make the analysis, the end-user has to add only the sample. This biosensor is characterized by a linear range comprised between 1 and 12 mM and a detection limit of 0.05 mM.

The porosity of the paper and the vertical microfluidics have been also exploited for the easy and accurate sampling of sweat to deliver a pump-free wearable device. Li et al. [31] designed an origami double enzymatic biosensor with a multi-layer structure to boost the sweat diffusion, the entrapment of the enzymes, and the accessibility of electrolytes for the detection of both glucose and lactate using glucose oxidase and lactate oxidase, respectively. Indeed, as designed, they vertically increased the area of the hydrophilic region for fostering the diffusion of sweat along the vertical direction into the paper substrate, through capillary-driven force (Figure 1A). This configuration allowed for avoiding fluid accumulation, ensuring that the biosensor was capable of detecting the biomarkers in the instant sweat rather than the accumulated sweat. In that way, the accuracy of instant detection of sweat composition is largely improved, furnishing the punctual concentration value, instead of a mean of values. The developed paper-based origami device demonstrated a dynamic range comprised between 0.08 and1.25 mM with a detection limit of 17.05 μM in the case of glucose biosensor, while in the range of 0.3–20.3 mM with a detection limit of 3.73 μM in the case of lactate biosensor, values useful to detect the physiological level of these biomarkers in sweat.

By exploiting the foldability of the paper, Wang et al. [32] designed a “pop-up” electrochemical paper-based analytical device inspired by pop-up greeting cards and children’s books for measuring the beta-hydroxybutyrate by simply re-adapting a glucometer and a commercial beta-hydroxybutyrate kit (Figure 1B). The pop-up structure enables the easy change of the fluidic path and the control of the timing for implementing the easiness of the analysis, displaying a good linear fit in the clinically relevant range of 0.1 to 6.0 mM (R^2^ = 0.96). The authors highlighted that the limit of detection of 0.3 mM is comparable with the limit of detection of commercial test strips, but the standard deviation is smaller in the latter case.

Contrariwise to biosensors able to detect the enzymatic substrate, the inhibitive biosensors have as the target analyte the enzyme inhibitors, and their electrochemical output is inversely proportional to the amount of the target analyte [56]. Furthermore, for the quantification of the inhibitor, the addition of the substrate for the measurement of the enzymatic activity in the absence and in the presence of the inhibitor is mandatory. The origami paper-based electrochemical biosensors allow for several smart configurations able to detect the inhibitors of enzymes by improving the simplicity of the measure.

Ding et al. reported [33] the first potentiometric origami paper-based biosensor using butyrylcholinesterase as enzyme and organophosphorus pesticides as target analytes, these pesticides being able to irreversibly inhibit this enzyme. In detail, the authors used as sensing layer a polymeric membrane designed for high sensitivity and selectivity toward butyrylcholine using heptakis(2,3,6-tri-o-methyl)-b-cyclodextrin as an ionophore. In the presence of enzymatic by-product choline, the sensor showed lower potential response than the butyrylcholine cation, and this ion-selective electrode membrane was selected for the detection of butyrylcholinesterase activity. Regarding the 3D structure, the origami biosensor was conceived with a test pad surrounded by three folding pads for loading the enzymatic substrate (substrate pad), the enzyme (enzymatic pad), and the sample (sample pad) (Figure 1C). The substrate pad was folded above the test pad and some μL of substrate solution was added onto the pad. Successively, the enzyme pad was folded and some μL of the enzymatic solution was spotted onto the enzyme zone. The sample pad was then folded on the top, and some μL of sample solution were added to the sample zone. Since it is an irreversible inhibition, for which an incubation time (reaction between the enzyme and inhibitor) increases the sensitivity, the sample pad and the enzyme pad were clamped by two binder clips to ensure a contact for 5 min. After that, the sample pad was unfolded, the inhibited enzyme pad was folded onto the substrate pad, and thus the enzymatic activity was measured by adding the phosphate buffer solution. In this proof of concept, the authors observed a dynamic range comprised between 0.1 and 1.0 nM and a detection limit of 0.06 nM using methyl parathion as target organophosphate. In this configuration, the presence of several pads allowed to avoid the different reservoirs for enzymatic reaction with the substrate and incubation time, making the whole process measurement suitable to be carried out in a miniaturized system.

The foldability of the paper and the easiness to be embedded as well as to cut the pads have been exploited by our group to design origami paper-based devices able to detect different classes of pesticides by using different types of enzymes able to be differently inhibited [10]. Furthermore, we exploited the porosity of the paper to load both the enzyme and the enzymatic substrate, delivering a reagent-free device, which requires the end- users to only add the sample or distilled water. This origami paper-based device was designed by integrating two different office paper screen-printed electrodes (placed back-to-back in the front and the backside of the origami system) with foldable filter paper strips, pre-loading different enzymes and enzymatic substrates on the latter. In detail, the analysis of different classes of pesticides, namely organophosphorus insecticides, phenoxy-acid herbicides, and triazine herbicide, was achieved using butyrylcholinesterase, alkaline phosphatase, and tyrosinase enzymes for the detection, respectively. For each class of pesticide, a filter paper strip was configured as follows: two pads were pre-loaded with the enzyme (red pads), and the other two pads were pre-loaded with the substrate (green pads) by simple drop-casting of the reagent solutions (A,C in Figure 1D). This configuration allows for the measurement of the enzymatic activity both without inhibitors and in the presence of inhibitors. In detail, when a distal couple of pads is folded to contact the electrochemical cell printed on the office paper, the non-inhibited enzymatic activity can be measured by the drop-casting distilled water. The choice of distilled water relies on the fact that the pads already contain the enzyme, its substrate and buffer salts, and the distilled water can dissolve them. For the sample measurement in the presence of inhibitors, some μL of water sample are loaded only onto the red pad containing the enzyme for the incubation time (5 min), then the pads are folded to contact the electrochemical cell printed on the office paper and the measure is recorded. The degree of inhibition was evaluated by chronoamperometric mode and the suitability with real samples was assessed by adding the river water samples without any treatment, thanks to the capability of paper to block the particulate, disclosing the feature of the paper pad of enabling the sample treatment. The paper-based device demonstrated the capability to quantify paraoxon, 2,4-dichlorophenoxyacetic acid, and atrazine at ppb level in both standard solutions and river water sample with good accuracy, demonstrating that exploiting the features of paper is possible to treat the sample, store the reagents, and make the measurement using a foldable paper-based device.

The porosity of the paper in the origami configuration was also exploited for the detection of the target analyte in the gas phase without any external sampling system, as demonstrated by our group [12] in the case of choline oxidase biosensors for chemical warfare agents, i.e., mustard agents, knowing that these compounds can reversibly inhibit the choline oxidase enzyme. In detail, an origami device constituted of two layers has been fabricated, one layer in which the electrochemical cell is printed, and the enzyme and phosphate salts are pre-loaded, and the second layer in which the substrate, i.e., choline, is pre-loaded, obtaining a biosensor ready-to-use without the necessity of additional reagents for the analysis (Figure 1E). The measurement was carried out by overlapping the two origami layers and exposing the biosensor to the mustard agents-containing aerosol. The porosity of paper allowed the layers of the origami to be impregnated by the aerosol and thus to dissolve the reagents, enabling the inhibition against the enzymatic reaction to be measured. This paper-based device was first tested with a toxic mustard agent simulant, (bis-(2-chloroethyl) amine), and subsequently with a real mustard agent in compliance with the appropriate security measurements at the Bundeswehr Institute of Pharmacology and Toxicology (Munich, Germany) in both solution and aerosol phase, reaching a detection limit equal to 1 mM and 0.019 g/m^3^ in the liquid and aerosol phase, respectively. This work highlighted that the paper-based device is able to overcome the constrain of the polyester printed sensors, that in the case of aerosol require an external engineering sampling system [57], resulting in the need for a more complex analytical device.

## 3. Origami Paper-Based Electrochemical DNA Sensors

Selective electrochemical biosensors are obtained by employing nucleic acid sequences as specific recognition elements. Biosensors based on nucleic acids have been widely explored over the last years due to their manifold physical, chemical, and biological properties [58]. Among them, aptasensors have been widely explored [59], basing on the use of aptamers, namely small artificial sequences of single-stranded DNA or RNA (ssDNA or ssRNA) capable of selectively recognize the specific target, including proteins, peptides, and various small molecules of analytical interest [60]. The key properties of nucleic acids rely on both the high specificity of the hybridization between complementary strands of DNA/RNA and the possibility to be conjugated to a sensing element (e.g., electroactive species, recognition elements), which confers to them an extreme versatility. During the last decades, the advances achieved by synthetic biology have converged the knowledge of genetic, chemistry, engineering, and biology to develop the technology for re-designing genes or proteins, taking inspiration from nature and going beyond it. The realization of synthetic antibodies, as well as synthetic nucleic acid molecules, have opened the way toward a wider horizon in the biosensor field. Indeed, nowadays nanotechnology allows for a fine design of synthetic nucleic acid molecules, providing tailor-made sensing materials for a variety of applications by using simple procedures and instrumentation. These advantageous features have significantly boosted the development of cost-effective nucleic acid-based biosensors, also reaching the field of paper-based sensors during the last years [61]. Several strategies have been explored for integrating nucleic acid molecules as sensing elements in the cellulosic network of paper, showing that the paper matrix is a suitable platform for realizing nucleic acid-based biosensors [62,63,64].

In the framework of origami sensors, Henry’s group contributed with their experience in paper-based sensors to realize a nucleic acid-based biosensor designed with an origami configuration [34]. The origami sensor was developed by covalently immobilizing a pyrrolidinyl peptide nucleic acid (acpcPNA) onto partially oxidized cellulose paper. In detail, the procedure employed consisted of a mild oxidative treatment, based on LiCl in NaIO_4_, on the cellulose matrix to generate aldehyde groups functional to the immobilization of the acpcPNA in the presence of NaBH_3_CN. The authors took advantage of the covalently binding of the acpcPNA to the cellulose network to build up a device that can be multiply regenerated by simply replacing the overlaying pad, avoiding any leaching of the acpcPNA. The nucleic acid probe was immobilized on the screen-printed working electrode, while the reference and counter electrodes were screen-printed on a second pad, to be overlapped for the measurement (Figure 2A).

The device was tested for the detection of mycobacterium tuberculosis as an example application. In detail, a synthetic 15-base oligonucleotide of mycobacterium tuberculosis was chosen as target ssDNA able to hybridize with the immobilized acpcPNA, obtaining an electrochemical DNA sensor with a label-free approach. The complex formation resulted in the variation of the charge transfer resistance, monitored by electrochemical impedance spectroscopy using [Fe(CN)_6_]^3−/4−^ as the electrochemical probe. The analytical performance of the developed origami was proved for the detection in real clinical samples confirmed by the comparison with PCR reference method, allowing for a linear response in the range from 2 nM to 200 nM of mycobacterium tuberculosis and limits of detection and quantification in the nanomolar level.

The origami folding was recently combined with the horizontal transport by capillary forces by Jiang et al. [35]. The authors have developed a microfluidic aptasensor on a single piece of chromatography paper patterned by polydimethylsiloxane, with microchannels and screen-printed electrodes (Figure 2B). The working electrodes were obtained separately from the counter and reference electrodes. The origami was designed with a symmetrical geometry, having two fold lines to be bent for completing two analogous electrochemical cells. This configuration allowed for a dual-target detection or screening at a time. The working electrode was electrodeposited with black phosphorus nanosheets decorated with an aptamer as the molecular recognition probe, being specific for the recognition of the peanut allergen Ara h1 without the need for labeling. The detection was carried out by using a ferro/ferricyanide redox probe in differential pulse voltammetry, achieving a good linear response in the range of 50–1000 ng/mL and a detection limit of 21.6 ng/mL.

A more sophisticated origami architecture was conceived by Crooks’ research group, who realized an origami that incorporates slip layers (oSlip) for the detection of DNA from hepatitis B virus (HBV) [36]. In detail, the “slip pad” consisted of a moveable layer of paper that can be used to initiate on-chip chemical reactions at the desired time. This component was provided with hollow channels for realizing a one-step device assembled in a sandwich-like configuration (Figure 2C). The layering of this device was exploited for selecting the paper layers when needed to load the reagents or to control the flow through the sandwich device.

The device exploits both silver nanoparticle and magnetic microbeads for obtaining an amplification of the response equal to a factor of 250,000 and ∼25-fold, respectively. The magnetic microbeads are used as mobile solid phase to support the capture probe, which was designed to bind on separated portions of the sequence both the ssDNA from HBV and the ssDNA labeled to the AgNPs. Paper allows for the concentration of these particles at the electrode, enabling signal amplification. In detail, the sample containing the magnetic microbeads-DNA-AgNPs, obtained after one-step hybridization, was injected in the inlet hollow and allowed to flow through the layers; as the solution reaches the bottom layer, it flows upward through the outlet hollow and solves a colorimetric indicator, revealing that the device is ready for the electrochemical assay. The reaction is then activated when the slip layer is moved to its functional position, enabling the dissolution of KMnO_4_ to oxide Ag contained in AgNPs to Ag^+^, which is detected in voltammetric mode with a detection limit of 85 pM. The presence of ∼250,000 Ag atoms in each AgNP explains the amplification of the signal resulting from this configuration.

To verify the performance of this origami, the voltammetric response was compared with a conventional electrochemical cell, reproducing the HBV-DNA sandwich assay. Interestingly, the response of the multilayer paper device was significantly higher than the response resulting from the conventional electrochemical cell. The authors attributed this improvement of the signal to the geometric confinement of Ag^+^ near the working electrode, ensured by the wax hydrophobic barrier that limits the possibility of diffusion of the oxidized Ag^+^. This effect is combined with the direct application of a magnet at the working electrode, which ensures the close proximity of the magnetic microbeads to the electrode.

A noteworthy study of the unique properties of origami-like devices was conducted by Ye et al. in 2018 [37], focusing on the advantages that the 3D vertical flow configuration of paper-based origami devices can provide. Indeed, they designed an origami geometry to realize a molecular threading-dependent transport system for the controlled and directional transport of biomolecules. The vertical configuration was conceived to enable molecular recognition and enzymatic reaction with a programmed order. Importantly, the authors drew attention to the hierarchical structure of the paper material, which is highlighted as a versatile and multitasking material among many examples in nature, having the mass storage capability and the capillary properties as key features.

The advantage of a 3D vertical configuration relies on the possibility to overcome the Lucas−Washburn law, which limits the mass transport distance to about 3−4 cm in 2D lateral flow devices. This was shown by comparing the transport of methylene blue between a 2D paper device, on which the diffusion was driven only by capillary forces, and a 3D paper architecture (obtained by folding the paper strip), observing a sharply greater coloring for the latter (Figure 2D).

After demonstrating the concept by visual mode, the authors have applied the origami device for multiple components storage and transport, namely DNA and proteins. In detail, biotin-DNA and avidin-HRP were dropped on dedicated areas of paper (preserving chips) and transported through the 3D configuration from a reaction area to a detection area (reaction and detection chips, respectively). The transport was carried out under controlled conditions, by programing the 3D folding of the paper device. A cascade reaction was triggered as soon as the biomolecule reach the reaction area, modified with AuNPs, where a capture probe was immobilized (ssDNA and tetrahedral DNA nanostructures). The foldability of paper was herein skillfully exploited to drive the cascade reactions also exploiting the porosity of paper to transport the regents to the desired site or, on the other hand, to store the immobilized receptors. Interestingly, such a complex design allowed for a rapid and single-step measurement of a target DNA molecule with picomolar detection limit and the capability of distinguishing a single base mismatch.

## 4. Origami Paper-Based Electrochemical Immunosensors

Immunosensors are analytical devices used to detect the binding event between an antibody (Ab) and an antigen (Ag) with the formation of a stable complex [65]. For immunosensor development, either Ab or Ag can be immobilized on the surface of different transducers, producing several immunosensor configurations with high sensitivity and selectivity thanks to the high specificity of antigen–antibody interactions [66,67,68].

Recently, paper-based immunosensors have been used as for the development of point-of-care testing kits exploiting the lateral flow assay (LFA) technology [69]. The LFA-based paper-based sensors generally consist of a sample pad, a conjugate pad, a nitrocellulose membrane, and an absorption pad. In these sensors, test analytes pass horizontally from the sample pad to the test section and the absorption pad by capillary force, allowing the binding between antigens and labelled antibodies on the conjugation pad for a sensitive detection. Although LFA-based electrochemical immunosensors have provided relatively short assay times (approximately 10–20 min), low-cost analysis, simple handling, and ease of mass production [70], it has some limitations such as relatively low sensitivity, limited sample volume, and difficulties in making multiple measurements [71].

In the last decade, three-dimensional microfluidic paper devices have been developed to improve the lateral flow-based sensors and to overcome their limitations leading to the development of paper-based vertical-flow immunosensor, which allows rapid vertical flow assay systems with controlled vertical flow and a separate measuring area [72]. For instance, Bhardwaj et al. [38] developed a vertical flow-based paper immunosensor using a different pore size sample pad for the electrochemical detection of the influenza virus H1N1 in both standard buffer solution and saliva samples. This lateral flow-based paper immunosensor consists of a different pore size sample pad (a pad characterized by pores with different sizes i.e., larger pores with a 11 μm diameter and smaller pores with a 0.45 μm diameter), a conjugate pad, a nitrocellulose membrane strip, and an absorption pad, all of which are vertically stacked one upon the other onto a polyester backing film. The three-electrode area was defined on a nitrocellulose membrane strip by wax printing (Figure 3A). The different pore size pad allowed them to increase the binding efficiency of antigen-horseradish peroxidase-tagged antibodies on the conjugate pad and concentrate the antigen–antibody complexes by providing the optimal residual time, fast detection (~6 min), and high sensitivity, with a limit of detection lower than 5 PFU mL^−1^ for saliva samples. Moreover, the porosity of sample pads acted not only as flow/volume control components, but also as a filter that facilitates small-sized biological particles such as viruses to pass through, while retaining larger particles, which is useful for the detection of complex fluids.

A paper-based electrochemical vertical flow paper-based device was also developed by Wang et al. [39], for the detection of 17β-E2 using multi-walled carbon nanotubes/thionine/gold nanoparticles composites synthesized and coated directly on the screen-printed working electrode for the immobilization of anti-E2. The device, which consisted of four layers, was made on four pieces of cellulose filter papers whose size was 10.5 mm × 35.0 mm. The samples were injected from the sample inlet, flowed through the microchannel and entered the filter hole. After filtering, they finally arrived at the reaction site. This vertical flow-based structure offers different advantages, including small volume of required samples, a simple procedure of sample handling, and an increased sensitivity, automated flow-injection and samples filtration.

As already discussed, the possibility to fold the paper to obtain different origami geometries allows for the fabrication of 3D structure without the need for other materials, e.g., double side tape, to integrate the different layers together. This strategy revealed to be advantageous also for the development of origami-based immunoassays. For instance, Li et al. [40] designed an immunosensing device for the detection of prostate antigen using two waxed pads, one for the screen-printed counter and reference electrodes and the other one for the working electrode, which were folded to create the 3D structure (Figure 3B). The detection of prostate antigen was carried out using an electrochemical enzymatic redox cycling constituted by glucose oxidase as an enzyme label, 3,3’,5,5’-tetramethylbenzidine as a redox electrochemical mediator, and glucose as the enzymatic substrate. The porosity of the paper was also exploited to grow gold nanoparticles directly on the surfaces of cellulose fibres in the working electrode where subsequently, manganese oxide nanowires were electrodeposited to form a network with large surface areas. The proposed method successfully fulfilled the highly sensitive detection of prostate antigen with a linear range of 0.005 ng/mL–100 ng/mL with a detection limit of 0.0012 ng/mL.

Another example of origami configuration was reported by Li et al. [41], who exploited the paper folding to create a biosensor able to detect HIV p24 antigen in human serum with a low detection limit of 300 fg/mL (>33 times lower than that of a commercial p24 antigen test kit), integrating hydrothermally synthesized zinc oxide nanowires and electrochemical impedance spectroscopic technique. In this case, paper was exploited for the in situ growth of zinc oxide nanowires directly on a carbon working electrode and then the zinc oxide nanowires were functionalized with p24 antibodies. The device consisted of two pieces of cellulose paper: (i) one piece of paper containing a hydrophilic paper test zone patterned via solid wax printing, and the carbon counter and the silver/silver chloride reference screen-printed electrodes, and (ii) another piece of paper including the carbon working electrode, on which zinc oxide nanowires are directly synthesized in situ (Figure 3C). The hydrothermal growth of ZnO NWs on the paper substrate included two steps: first, an uniform coating on the paper with a seeding layer of ZnO nanoparticles, which provides the starting points of the ZnO-NW growth; then, the directional nucleation of ZnO NWs from the seeding layer. In this second step the ZnO-NP-coated paper was immersed in an aqueous solution of zinc salt and other chemicals at an elevated temperature for the growth of ZnO NWs. The authors chose the origami structure and arranged the working electrode on a piece of paper separated from the origami paper layer for two main reasons. First, the zinc oxide nanowires on the working electrode have intimate contact with the solution contained in the test zone, yielding enhanced electrochemical performance; Moreover, the hydrothermal growth and surface biofunctionalization of zinc oxide nanowires involve heating and immersion in solutions which could compromise the hydrophobicity of wax barriers if the working electrode was printed directly on the origami paper layer.

The same approach was used by Reucha et al. [42] for the development of an origami electrochemical platform for sensitive detection of human IFN-γ. Indeed, the origami consisted of two wax patterned separated pads (Figure 3D). The working electrode was designed separately from the counter and reference electrodes to reduce the consumption of the reagents and the sample volume, as well as to prevent reference and counter electrodes contamination with proteins during the preparation of the immunosensor. For the detection of human IFN-γ, a monoclonal human IFN-γ antibody was immobilized on the polyaniline modified graphene screen-printed paper electrode and electrochemical impedance spectroscopy was used for detection of human IFN-γ in a range of 5–1000 pg/mL with a detection limit of 3.4 pg/mL.

The possibility of designing immunosensors with more complex configurations has been achieved by folding different layers of paper to realize one origami structure for the simultaneously and selectively detection of different analytes. Indeed, Sun et al. [43] proposed an origami multiplexed enzyme-free electrochemical immunosensor for the detection of human chorionic gonadotropin, prostate-specific antigen, and carcinoembryonic antigen. The electrochemical immunosensor is characterized by zinc oxide nanorods which provide high number of sites for conjugating the capture antibodies and reduced graphene oxide which improves the electronic transmission rate. The current signal is generated from the reduction of H_2_O_2_ and further amplified by a subsequent signal labels-promoted deposition of silver. The origami device was comprised of an auxiliary pad surrounded by three sample pads of the same size (20 mm × 20 mm). The electrode array consisted of a screen-printed Ag/AgCl reference electrode and a carbon counter electrode on the auxiliary zone and three screen-printed carbon working electrodes on the three paper sample zones, respectively, which are functionalized with three different capture probes for their specific analytes. Between each pad and auxiliary pad, an unprinted line (1 mm in width) was defined as a fold line which ensures that the paper sample zones on the three pads were properly and exactly aligned to the auxiliary pad after folding. Under optimal conditions, the proposed immunosensor exhibit excellent precision, high sensitivity, and a detection limit of 0.0007 mIU/mL for human chorionic gonadotropin, 0.35 pg/mL for prostate-specific antigen, and 0.33 pg/mL for carcinoembryonic antigen.

Another example of multiplexed 3D electrochemical immunosensor is proposed by Shen et al. [44] with a label-free field-effect transistor/chemiresistor-based immunosensor. This sensor consisted of pyrene carboxylic acid-modified single-walled carbon nanotubes deposited by quantitative inkjet printing with an optimal three-dimensional semiconductor density on the paper substrate. Monoclonal anti-human antibodies were individually immobilized onto the SWCNTs surface to achieve a highly sensitive and specific detection of human serum albumin and human immunoglobulin G with detection limit of 1.5 pM. The origami biosensor composed of five-petal shaped hydrophilic channels on paper was designed to equally split one sample into five aliquots for individual sensing channels by the capillary force (Figure 3E). In details, the origami was composed of three layers: (1) top layer for sample-splitting and paper-bridging; (2) middle layer with chemiresistor biosensor arrays for multiplexed detections; and (3) bottom layer with sufficient absorbing capability. Unlike the manually assembled ones, origami devices benefited from the precise trimming by CO_2_ laser cutting that helped facile alignment of the three layers thereby reducing the human labor and error.

In alternative to the foldable configurations, Crooks’ research group applied the approach of the slipping pads, already discussed for the DNA-based origami biosensors [36], also in the application field of origami-inspired immunosensors, aiming to further improve the possibilities of designing reconfigurable structures of origami devices. They proposed two different origami slip pads (*o*SLIP) where the device is fabricated by paper folding and operated by paper slipping. The first paper-based immunosensor is based on quantitative detection of silver nanoparticle labels linked to a magnetic microbead support via a ricin immunosandwich with a detection limit of 34 pM [45]. The sensor platform comprised four wax-patterned paper layers. The three carbon electrodes were stencil-printed on the lower layer 1, which displays the inlet and the outlet reservoirs. Layer 2 contained a hollow channel and a paper reservoir loaded with a blue dye, the latter indicating that the device was ready for measurement as soon as the colouring occurred (due to the sample flow through the default path within the microfluidic channels). Layer 3, namely the slip layer, contained both a hollow channel and a paper pad for dried oxidant storage. Finally, layer 4 consisted of a hydrophilic layer (hemichannel) and a sink pad that drove a continuous flow of fluid through the device until its capacity is filled. The oSlip is firstly assembled by folding the paper. The assay began by injecting the pre-formed ricin immunocomposite into the oSlip inlet. Then, the ricin immunocomposite is concentrated under the first carbon electrode by the magnetic field. When the blue colour appeared due to the flow reaching the dye at the outlet, the pre-dried chemical oxidant was slipped (by pulling Layer 3 until a green indicator line became visible) reaching a direct contact with the ricin immunocomposite. Finally, the dissolved Ag^+^ ions are electrodeposited on the electrode as metallic Ag for 200 s and then stripped off. This work has demonstrated how the combination of folding components and layers that can be slipped into and out of the origami can provide sophisticated configurations that allow for fine control of the different steps for the device application.

The second origami slip immunosensor structure was used to detect the kidney disease marker Trefoil Factor 3 (TFF3) in human urine [46]. The sensor is based on a quantitative metallo-immunoassay able to determine TFF3 concentrations via electrochemical detection of environmentally stable silver nanoparticle labels attached to magnetic microbeads via a TFF3 immunosensor. For the electrochemical detection, a one-step assay was performed characterized by incubating TFF3, the AgNP/2°mpAb/mpAb conjugate, and the microbead/pAb conjugate simultaneously. After this single step, the immunocomplex was washed three times and then it was injected into the inlet of the oSlip for the measurement. Similarly, to the work previously described [64], this device design is characterized by a hollow channel and a hemichannel, which enable the microbeads to flow rapidly through the origami, and a slip-layer switch, which allowed for time-controlling of the reagent delivery. Moreover, the *o*Slip configuration allows for the preconcentration of all necessary reagents making easier the work of end-user and allowing for the detection of TFF3 in human urine in the concentration range comprised between 0.03 and 7.0 μg/mL.

## 5. Origami Paper-Based Electrochemical MIP Sensors

Molecular imprinting is a powerful technique based on a mimic approach, inspired by natural receptors to achieve molecular recognition [73]. This strategy aims to reproduce antibody-like binding properties or enzyme-like catalytic activities by using a polymeric template markable with the target analyte, in order to leave an imprint that can ensure a selective and specific recognition. Usually, MIPs are obtained by the cross-linking of functional monomers, which are polymerized under specific conditions in the presence of template molecules by covalent, non-covalent, or hydrogen interactions [74]. After the polymerization, the template molecules are removed from the polymer, thus leaving cavities that are able to act with a key-lock mechanism when the template molecules are detected in a working sample [75]. This approach allows for exploiting the molecular memory imprinted in the polymer by the target molecule itself, which is first used as a template and then ensures the high affinity in shape and size at the measurement stage.

MIPs have been used for a variety of applications ranging from chromatographic separation to molecular sensing [76,77], but also serving as a tool for drug delivery [78]. In the field of electrochemical sensors, they are known as smart modification materials characterized by short synthesis time, largely improved storage stability, and cost-effectiveness. One of the main challenges for this technology is the imprinting of biomacromolecules, such as proteins, because of their complex structures and large sizes. Moreover, the use of MIPs for electrochemical devices is typically affected by low sensitivity due to the poor conductivity and electrocatalytic activity of most of the common polymers employed. In order to overcome these limitations, various approaches have been explored, including strategies to improve the surface imprinting efficiency or the use of conductive nanomaterials.

Over recent years, the paper was showed to be a suitable platform for the in-situ synthesis of MIPs, and its properties have contributed to the realization of MIPs, such as the porosity and the adsorptive cellulosic network [79,80]. Also in this field, the versatility of paper has disclosed new possibilities for expanding the applicability of MIP technology.

Very recently, the MIP technology was integrated with the benefits carried by origami structured paper devices. Amatatongchai et al. [47] applied a simple origami configuration to develop a paper-based biosensor for serotonin detection. The biosensor was based on a graphite-paste electrode modified with a MIP composed of Fe_3_O_4_@Au nanoparticles encapsulated with imprinted silica. Circular hydrophobic areas were realized on filter paper by alkyl ketene dimer inkjet printing. The three-electrode cell was screen-printed using a graphite paste comprising graphite powder, carbon nanotubes, and mineral oil in correspondence with the hydrophilic areas. After printing the conductive paste on a single piece of paper, the Fe_3_O_4_@Au@SiO_2_-MIP nanocomposite was drop-cast onto the working electrode. Thus, the sensor was ready to be folded, overlapping the two circular hydrophilic areas, and to be used for the measurement (Figure 4A).

The configuration of this origami device was further studied by varying the number of layers overlapped and by evaluating the effect of increasing volumes of the sample. As introduced above, each paper layer presented a circular hydrophilic region delimited by the alkyl ketene dimer hydrophobic barrier. After vertically aligning the hydrophilic areas, a serotonin sample was drop-cast on the hydrophilic region on the top of the origami sensor. An increment of the voltammetric response was observed when a second paper layer was overlapped on the underlying screen-printed paper layer, while the addition of a third paper layer did not change the sensor response significantly. Although this evidence would be worthy of further investigation, the authors did not provide a critical explanation and continued the study by choosing the double-layer configuration. Regarding the effect of the sample volume, the increment from 10 μL to 20 μL resulted in increasing the signal recorded. The authors ascribed this behavior to the facilitated diffusion of the sample through the layers when using a larger volume. However, they highlighted that a further increase of the volume can be responsible for an overload of the sample and the eventual occurrence of noise due to wetting of the electrical contact. Importantly, the authors pointed out how the use of paper layering to design a folded origami configuration was capable of avoiding the direct contact of the sample with the electrode surface, especially important for the real samples having complex matrices to reduce the possible interfering effects on the sensor performance. Moreover, the high surface-to-volume ratio offered by the paper-fiber matrix allows for a high impregnation of the sample within the sensor material, favoring the detection. The origami sensor modified with the Fe_3_O_4_@Au@SiO_2_-MIP nanocomposite was proved to be suitable for the detection of serotonin in pharmaceutical and urine samples, with good analytical performances in terms of precision and high tolerance toward interfering chemicals.

Further advances in the application of MIP to paper origami biosensors were achieved by Yu’s group [48], who proposed an ultrasensitive sensing platform for the rapid and accurate detection of glycoproteins. The origami device was composed of three regions: a detection pad, a channel pad, and a washing pad. The electrodes were screen-printed on the detection pad, with the working one printed on the reverse side. A circular area (8 mm in diameter) was placed in correspondence with the working electrode but on the obverse side. A smaller circular area (5 mm in diameter) was placed laterally and used as an inlet zone for the sample. The channel pad was overlapped on the detection pad by alternatively folding along two fold lines to switch the sensor from the washing mode (fold line 1) to the detection mode (fold line 2). Using fold line 2, the channel pad allows the sample to flow from the inlet zone to the electrodes, thanks to the capillary forces through the main hydrophilic channel, thus enabling the electrochemical measurements. For the washing step, a buffer was added in the 8-mm circular pool to wash the detecting area. A semi-hydrophilic channel was designed on the channel pad to draw the excess of washing solution by capillary forces, driving it toward the washing pad. The semi-hydrophilic channel was conceived to further protect the detection area from being contaminated by the washing liquid (Figure 4B).

A glycoprotein-based MIP was synthesized on the working electrode after the in-situ growth of Au nanorod in conjunction with 4-mercaptophenylboronic acid as a recognition element, able to form a covalent cyclic ester with the target glycoprotein. The detecting principle relied on AuNPs immobilized on the surface of SiO_2_ and functionalized with 4-mercaptophenylboronic acid nanoparticles able to link DNA molecules, obtaining a SiO_2_@Au/dsDNA/CeO_2_ nanocomposite. Double strands of DNA were allowed to form on the surface of SiO_2_@Au through a hybridization chain reaction in the presence of two hairpin DNAs. Thus, CeO_2_ nanoparticles were bound to the DNA probes through an amidation reaction. The target glycoprotein, namely ovalbumin (OVA), was recognized by the boronate affinity-based MIP, enabling the sandwich interaction with the SiO_2_@Au/dsDNA/CeO_2_ nanocomposite. The detection was hence obtained by the reduction of Ce^4+^ to Ce^3+^ upon the addition of 1-naphthol reagent. In this configuration, the SiO_2_@Au/dsDNA/CeO_2_ nanocomposite served for obtaining higher electron transfer efficiency and larger surface area for the DNA immobilization at the same time.

The morphological and electrochemical suitability of this origami device was analyzed in detail. Micrographs of the chromatography paper revealed a framework of cellulosic fibers that was responsible to provide a biocompatible, incompact microenvironment for the in situ growing of Au nanorods. The successful formation of the MIP was verified as well. The comprehensive characterization of the electrochemical features of the Au-μPADs/MIPs as well as the promising analytical performance achieved with standard samples enabled the authors to apply the origami device to real samples, consisting of 100-fold diluted egg white samples, resulting in high analytical accuracy.

A sophisticated example of the integration of microfluidic properties of paper-based analytical devices with the surface bio-molecularly-imprinted technique was reported by Qi et al. [49] for the selective and sensitive clinical detection of carcino-embryonic antigen. Interestingly, a movable valve and circular rotating pieces of paper were used to allow for a modulable control of the device. This bio-molecularly-imprinted device was designed to carry out antibody-free biomarker analysis by in-situ synthesized MIP. This strategy allows for the direct detection of antigens avoiding the issue of the antibody preservation typical for enzyme-linked immunosorbent assay.

The origami configuration was conceived for applying a controlled overlapping of the sample areas and the working electrode area upon folding the origami. The use of a movable valve was chosen to realize a configuration suitable for a multi-step electropolymerization process, achieving satisfactory results also over long times of polymerization (∼1 h). Moreover, the possibility to rotate the circular parts, on which the counter and reference electrodes were printed, was used to allow the electrochemical cell to be completed both before and after the origami folding, depending on the need.

In detail, four different paper components were designed to fabricate the origami device: (i) the working electrode part hosting two working electrodes; (ii) the counter/reference electrode part, composed of four independent rotating circular pads; (iii) the washing part, with two washing channels; and (iv) the movable valve. The latter was connected to the working electrode part by means of rivets and was placed in its functional position when necessary. The working electrode part and the washing part were obtained on the same piece of paper and designed to be overlapped by folding the origami. The washing part presented hydrophobic channels and two waste pools for transporting waste solution (Figure 4C).

The origami biosensors were prepared by surface imprinting technology, exploiting the sophisticated architecture of the origami device. Two identical sets of reaction units, each one configured as described above, were available on the same device in a symmetrical geometry, one of which was served as a blank baseline. A sequential modification of the working electrode with graphene oxide, chitosan and glutaraldehyde was carried out with the origami unfolded. When carcino-embryonic antigen was added as a molecular template, the counter/reference electrode parts were rotated to overlap with the working electrode areas, allowing for completing the electrochemical cell. Dopamine was added into the synthetic material pool, close to the working electrodes but separated by hydrophobic regions; thus, the movable valve was moved to connect the pool to the working electrode and let dopamine flow to reach the cell for the electropolymerization process by cyclic voltammetry. Hence, the origami was folded, to overlap the working electrode areas with the pools present on the washing area, and the dropping of an eluent allowed to remove the excess of template carcino-embryonic antigen.

The rotating parts of this origami were exploited also for the measurement of standard and real samples. Firstly, keeping the origami unfolded and the circular parts and the valve away from the working electrode area, the samples were loaded on the working electrode parts for a resting period, during which the analyte diffused inside the MIP structure. Thus, the origami was folded in order to overlap the washing part with the working electrode areas and let the excess sample solution being removed by a washing step. Finally, the circular part was rotated to overlap the counter and reference electrodes on the working electrode, and the voltammetric detection was performed.

The performances of such complex origami were carefully interrogated by electrochemical studies. Importantly, the suitability of the movable valve to allow the dopamine to flow to the working electrode area was successfully verified by comparison with direct drop-casting on the working electrode. After a comprehensive characterization of the surface molecularly imprinted process, the analytical performance of the Bio-MIP-ePADs was determined, obtaining satisfactory analytical features in terms of sensitivity, selectivity, and accuracy.

## 6. Origami Paper-Based Electrochemical Cell-Based Biosensors

The use of cells as biosensing elements for electrochemical biosensors turns to be necessary to obtain direct information about the cytotoxicity of chemicals as well as to monitor the inter-related effects of cytotoxicity on cell physiology. As is well known, the toxic effects occurring into living cells can trigger a cascade of reactions typically involving several components of the cell, thus requiring an in-situ and/or in vivo monitoring for understanding the overall consequences.

Various types of cyto-sensors, including chemiluminescent, fluorescent, electrochemical, electrochemiluminescent, and surface-enhanced Raman scattering cyto-sensors have been developed. Especially in the biomedical field, the sensitive and selective detection of cancer cells is fundamental to provide an early diagnosis and to plan a proper therapy.

Recently, the use of paper has been introduced as a novel platform for cytological or histological studies. Paper-based cyto-devices have begun to stand out in the field of sensors offering the attractive potential of transporting the cytological/histological research and applications on versatile, simple, miniaturized, and low-cost devices with an in-situ and rapid read-out. Firstly, the analytical methods applied for the study of paper-based cyto-devices exploited the colorimetric approach and techniques of fluorescence imaging. The development of this sector also in the field of the electrochemical sensor will provide an important contribution to multiple areas, including basic scientific advancement, clinical diagnostics, therapeutics, and energy storage/production.

Few studies have been reported about electrochemical cyto-devices with an origami principle. A first example was conceived by Su et al. [50] for obtaining an in vivo-like cell culture for human acute promyelocytic leukemia cells (HL-60), used as a proof-of-concept. Cell-targeting aptamers, namely KH1C12, were chosen as molecular probes for specific recognition of HL-60 cell.

The device was fabricated on a single sheet of flat paper, divided into two main squared areas: a paper cell pad (red square) as the working electrode area, and a paper auxiliary pad (blue square) with the counter and reference electrode (Figure 5A). The former was composed of four circular paper cell zones defined by the wax pattern, while the latter presented a single circular hydrophilic zone. The two areas were supposed to be perfectly overlapped upon bending on the folding line, thus completing the paper electrochemical cell.

The preparation of the cyto-origami device was carried out through the immobilization of the biocomponents on the paper cell area, previously functionalized with AuNPs to improve the conductivity and ensure the immobilization process. In detail, the conductive paper cell zones were modified with KH1C12 aptamers, capable of immobilizing a homogeneous HL-60 cell suspension. For the measurement, the paper auxiliary pad is folded on the paper cell pad and clamped within two homemade circuit boards. The detection process starts with the loading of the horseradish peroxidase-labeled folic acid, used as electrochemical bio-probes into each paper cell zone, followed by the addition of o-phenylenediamine and H_2_O_2_ through the paper auxiliary pad. The folate receptor is specifically recognized by the previously immobilized HL-60 cell and enables the amplification of the electrochemical signal thanks to the labeled horseradish peroxidase enzyme, which catalyzes the oxidation of o-phenylenediamine by H_2_O_2_. The enzymatic activity is finally recorded produce by differential pulse voltammetry.

This cyto-origami device was tested for anticancer drug screening, by monitoring the apoptosis induced by the action of the anticancer drug. In detail, cell apoptosis is associated with the translocation of the membrane phosphatidylserine from the inner to the outer side of the plasma membrane. In this case, horseradish peroxidase-labeled annexin-V was used because it was able to interact with the phosphatidylserine, available only when cell apoptosis occurs. This study showed further advantages of exploiting the 3D culture of cancer cells in the paper cell areas.

Shortly after, Ge et al. [51] continued the study on leukemia cell lines by developing an origami paper-based device for the detection of K-562 (Figure 5B). The origami structure exploited in this case was similar in principle to the one above discussed. The electroanalytical performance of the sensor was improved by using a nanocomposite based on AuNPs, graphene, and an ionic liquid, while the cells were captured by specific binding with concanavalin A (Con A) and cell surface mannose. Phorbol 12-myristate-13-acetate was used to stimulate the endogenous generation of H_2_O_2_ in the cells, to be detected by voltammetric detection. The authors reported a promising detection limit for cell concentration, equal to 200 cells/mL.

Another example of a cell-based origami biosensor with an analogous configuration (Figure 5C) was recently developed by Jiang et al. [52]. The authors wanted to mimic the physiological conditions directly on the paper environment to realize a biosensor for the recognition of food allergens using mast cells. In detail, casein was chosen as the major allergen in cow’s milk. A composite based on graphene and a hydrogel was used to enhance the conductivity of the paper working electrode, where the Rat basophilic mast cells were immobilized by simple drop-casting. The detection of casein was performed by differential pulse voltammetry allowing the authors to measure this allergen with a limit of detection of 32 ng/mL.

## 7. Self-Powered Origami Paper-Based Electrochemical Biosensors

Among the emerging challenges in the sensor field and beyond, there is a need to develop self-powered devices able to operate independently, wirelessly, and sustainably. A substantial breakthrough in the sensor power supply concern occurred when the concept of self-powered sensors was first proposed in 2006 [81]. Such systems consist of sensors devised to generate an electric signal when mechanically or chemically activated, without the need for an external power source [82]. Self-powered devices have been proposed as clean energy harvesters, with the ambitious purpose to create autonomous tools that can harvest the energy from the environment to power themself, rather than from conventional power supplies [83].

This achievement would allow revolutionizing the fields of sensing, data transmitting, data processing, energy harvesting, and energy storage. In the framework of electrochemical sensors for analytical applications, self-powered wearable electronics would provide on-site and continuous monitoring of biomedical parameters, representing a significant advance for the monitoring during therapies and early diagnosis.

The researchers’ efforts are now dedicated to investigating flexible, miniaturizable, and versatile substrates for the development of efficient self-powered devices. Paper has been again highlighted as a promising platform material for providing significant advances in this field [84].

One of the very first examples of self-powered paper-based biosensors devised with origami characteristics was reported by the research group of Crooks [53]. The authors developed a self-powered biosensor able to generate a current without the need for an external power source, realizing a platform in principle adaptable to a range of target molecules, including antibodies, DNAzymes, or aptazymes.

The whole sensor was printed with a symmetric geometry on a single layer of paper (Figure 6A). To avoid the proximity of the printed conductive inks to the microfluidic pattern, the paper was divided into two sides supposed to be folded for the measurement. On one side, an inlet area was fabricated by wax printing, from which a pair of identical channels was originated. The channels recombined at the end of their pathways in a region destined to form two half electrochemical cells. On the second side, two electrodes were obtained by screen-printing of conductive carbon ink. When the paper is folded at the predefined fold line, the electrodes were overlapped on the region where the channels recombine, completing the two half electrochemical cells. After reagents preloading and drying, the device is folded and sealed by lamination, which serves also for connecting the device to copper wires for electrochemical connection. This strategy allows the device to retain the sample avoiding losses due to evaporation or other alterations, without the need for adhesives that can cause contamination or nonspecific adsorption.

During the preparation of the biosensor, biotin-labeled aptamers immobilized on streptavidin-functionalized microbeads were entrapped into the initial part of the channels. The detection principle relied on the immobilized aptamer as a recognition element for adenosine, chosen as the target analyte for a proof-of-concept study. The adenosine binding caused the release of a GOx-labelled ssDNA, which was able to catalyze the oxidation of glucose, finally resulting in the conversion of redox probe [Fe(CN)_6_]^3−^ to [Fe(CN)_6_]^4−^. The difference in concentrations of the redox probe between the sensing half-cell and control half-cell results in a voltage that is accumulated. In this way, the sensor behaves similarly to a battery, able to charge a capacitor. By connecting the system to a digital multimeter, the accumulated capacitor is immediately discharged giving a current read-out. The level of accumulated voltage was proved to increase linearly with an increasing concentration of glucose oxidase. The origami biosensor provided detection of adenosine up to 250 μM, with a detection limit of 11.8 μM. The authors highlighted that the integration of the capacitor for obtaining the accumulation of the concentration-dependent voltage allowed for increasing the sensitivity of the assay 15.5-fold in comparison with the direct measurement of the current.

Later on, Ge et al. realized an origami device that combined a dual chemiluminescence-photoelectrochemical detection principle with a paper supercapacitor, resulting in the collection of the generated photocurrent [54]. The sensor was obtained on wax-patterned paper shaped with a particular geometry. A reaction area (reported in red in Figure 6B) was bordered with a collection area (reported in blue in Figure 6B), from which two rectangular “legs” was originated (reported in green in Figure 6B). Hydrophilic circular pools were present on the reaction and collection areas, with a screen-printed working electrode and counter and reference electrodes, respectively, conceived for being overlapped and allowing for the formation of an electrochemical cell. The wax-patterned hydrophobic legs were designed to be diagonally folded and overlapped, providing the paper supercapacitor. This component was electrically connected to the reference electrode, being able to be charged after the chemiluminescence-photoelectrochemical reactions occurred. The supercapacitor was provided of drawn thin-film graphite electrodes that were soaked with H_2_SO_4_−PVA gel electrolyte and left to solidify.

The origami biosensor was adapted for the detection of adenosine triphosphate in human serum samples as a model example, using a specific binding aptamer for molecular reorganization. The principle of the assay relied on the use of N-(aminobutyl)-N-(ethylisoluminol)-functionalized gold nanoparticles ABEI−AuNPs−H_2_O_2_ as a chemiluminescence-inducing system and p-iodophenol as a chemiluminescence enhancer. Thioglycolic acid-capped water-soluble cadmium sulfide nanoparticles were employed as the active species to generate photocurrents, being capable of absorbing the chemiluminescence emitted by the ABEI−AuNPs− H_2_O_2_ system thus generating the photocurrent. The system is triggered when ATP is present as a target molecule, able to form a complex with two ssDNAs separately immobilized on both the AuNPs and the CdS NPs, which draws the AuNPs sufficiently close to the cadmium sulfide nanoparticles enabling the chemiluminescence-photocurrent system. H_2_O_2_ was used as a sacrificial electron donor to provide the electrons for the valence-band holes and complete the photocurrent generation cycle. The paper supercapacitor allowed for the storage of the photocurrent produced (for a period of 180 s), which was amplified and recorded as soon as the device was connected to a digital multimeter.

The porosity and large surface area of the cellulose fibers were exploited to in-situ modified the working electrode by assembling poly(dimethyldiallylammonium chloride) and carbon nanotubes, which enhanced the conductivity of the paper and provided fast electron-transfer to ensure the efficiency of the photoelectrochemical signal through the chemiluminescence-inducing system. Indeed, the comparison of the developed configuration with a device obtained on ITO support revealed that the role of the modified paper was determinant for the resulting performance of the sensor, enhancing the response of about 3-fold in intensity.

An interesting example was given by Yu’s research group, who demonstrated that the self-powered technology on paper can be successfully integrated with origami devices by designing a pop-up mechanism [55]. The resulting biosensor provided dual-mode read-out, exploiting both direct voltametric measurement and the supercapacitor mode. The two read-out modes were designed for ATP detection as a model analyte. The pop-up structure was a key feature to control the microfluidic pathways of the device as well as the incubation times of the assay. The device was composed of an electrode area, the detection zone, the reaction zone, the hollow zone, and the supercapacitor system. The reaction zone and detection areas were spatially separated by a hydrophobic pattern, allowing the operator to connect them at the desired moment. Indeed, the detection area could be overlapped on the reaction area by applying a mechanical pressure in the clockwise direction, so that the working electrode and the carbon wire were put into contact.

During the preparation of the assay, the origami was kept unfolded. The hydrophilic paper in correspondence of the working electrode (detection zone) was modified with AuNPs to improve the conductivity as well as to allow the specific ssDNA and the aptamer to be sequentially immobilized by the affinity of thiol groups with gold. The aptamer was designed to undergo hybridization with the first ssDNA immobilized, and on the other hand to hybridize with a second ssDNA molecule labeled with glucose oxidase enzyme. Finally, ATP was incubated in the detection area. In the meanwhile, potassium ferricyanide and glucose were loaded on the reaction zone; moreover, the paper supercapacitor was obtained in the unprinted area of the origami by the deposition of graphite and PVA/H_3_PO_4_ electrolyte.

At the moment of the detection, the detection zone was overlapped with the reaction zone by exploiting the pop-up structure. The GOx-labelled ssDNA was released owing to the capture of ATP by aptamer and transferred to the reaction zone by vertical microfluidic, thus catalyzing the oxidation of glucose. The presence of pre-loaded ferricyanide acted as an electrochemical probe to acquire a DPV signal. Importantly, the concentration of species in the reaction zone resulted in the accumulation of a voltage that charged the paper supercapacitor. The connection of the system to a digital multimeter allowed for the acquisition of the resulting self-generated current. The origami biosensor was able to detect ATP concentration ranging from 10 to 5000 nM both in the direct voltametric measurement and the supercapacitor modes, with a limit of detection in the nanomolar level.

Another worthy of note approach is focused on the exploitation of microbial cells that can be exploited to produce green electric power. Indeed, the microorganisms’ capacity to catalyze the degradation of the organic compounds, such as from organic waste material, represents a sustainable approach for generating clean energy, particularly promising for those environments having limited energy sources. In detail, the physiological metabolism of microbial cells can be converted into reusable energy by transferring the resulting electron exchanges to an external electrode. In light of these attractive features, the integration of microbial fuel cells with microfluidic paper-based devices represents an ambitious purpose that could bridge the gap between this fascinating green approach and its practical realization.

A first attempt in this direction was made by Mohammadifar et al. [85], who built up a foldable origami device based on microbial fuel cells with the interesting capability of rapidly generating power with a small amount of bacteria-containing liquid, thanks to the rapid adsorption provided by the paper platform. An initial 2D geometry was obtained on paper, divided into four functional foldable areas, namely the anode pad, the reservoir pad, the cation exchange membrane, and the cathode pad (Figure 6D). The paper was patterned with wax to delimit hydrophilic areas. The cation exchange membrane was characterized by hydrophobic properties and proton/cation conductance. An anodic material was applied on the middle pad (i.e., the anode pad), which presented an inlet hole for injection of the bacterial liquid. The left tab was used as a paper reservoir to store the bacterial cells and the endogenous organic fuels. The origami was obtained by folding the tabs along the folding lines and the layers were kept together with an adhesive spray. After the bacterial cells were inoculated through the inlet hole, the protons/cations produced during the microbial metabolism diffused through the hydrophobic cation exchange membrane toward the cathode, reacting with oxygen. The authors investigated several composites with anodic properties to optimize both the bacterial attachment and the electron transfer, drawing attention to the dependence of the output voltages on the choice of the modifying material applied on paper. This study provided new insights into the construction of self-powered paper-based biosensors and further disclosed the perspectives for paper origami devices. Importantly, it shows how paper can be designed to create cathodic and anodic regions by exploiting different materials and a programmed folding. Moreover, the use of microbial cells and power generation on paper can play a significant role in future advances towards self-generated sustainable devices.

## 8. Conclusions and Perspective

The development of electrochemical paper-based biosensors has been started ca. 10 years ago by replacing the plastic-based support with the paper used as a substrate to print the electrochemical cell as well as microfluidic patterns with the overriding goal to deliver a pump-free microfluidic device characterized by an easy and cost-effective fabrication. The several features of the paper in the time have been exploited step by step, by fabricating analytical devices with additional features. For example, the porosity of the paper has been successively exploited to store the reagents for delivering a reagent-free biosensor or to treat the sample by simply adding the sample at the opposite of the layer in which the electrochemical cell is printed. The foldability of the paper has been recently used to design a 3D structure system to avoid the operator in performing multistep analysis, to control the timing, to reduce the sample manipulation, and to implement the multi-analysis. For instance, in the case of the origami paper-based devices for multiclass pesticide detection, the use of multiple pads combined with office-paper printed electrochemical sensors allowed to detect of three different classes of pesticides by only folding the pads, adding the distilled water, and cut the used pads used after the measure. In the case of the complex sample as whole blood, the use of several layers for vertical microfluidics combined with printed electrochemical sensors delivered an easy-to-use analytical tool for the application of precision medicine. Furthermore, the possibility to use the origami system with different layers loaded with different reagents can create an embedded device ready for the detection of the target analyte in the gas phase. Another additional value is related to the low cost of the paper rendering the device suitable for a single use, and thus overcoming the fouling problem. At the state of the art, the paper-based devices need to be industrialized by manufacturing them at the large scale. In addition, to fill in the gap between the bench and the market, a robust folding and unfolding procedure needs to be addressed. In the next future, we are confident that the recent efforts in printed electronics will pave the way for a new direction in origami paper-based electrochemical devices, with the final aims to fabricate in the next future integrated fully printed analytical tools, able, with the only addition of the sample, to automatically treat the sample, to contain the reagents, and to detect the signal.

## Figures and Tables

**Figure 1 biosensors-11-00328-f001:**
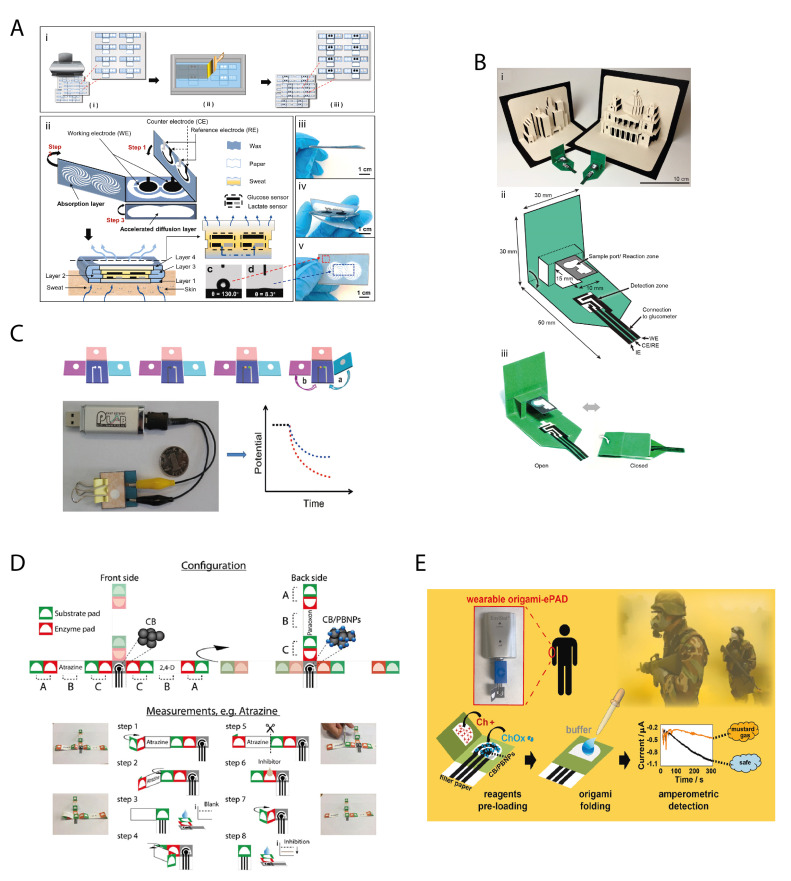
(**A**) A highly integrated sensing paper for wearable electrochemical sweat analysis. Reprinted with permission from [31], 2020 Elsevier; (**B**) a paper-based “pop-up” electrochemical device for analysis of Beta-Hydroxybutyrate. Reprinted with permission from [32], 2016 American Chemical Society; (**C**) a three-dimensional origami paper-based device for potentiometric biosensing. Reprinted with permission from [33], 2016 WILEY-VCH Verlag GmbH & Co; (**D**) origami multiple paper-based electrochemical biosensors for pesticide detection. Reprinted with permission from [10], 2018 Elsevier; (**E**) a wearable origami-like paper-based electrochemical biosensor for sulfur mustard detection. Reprinted with permission from [12], 2019 Elsevier.

**Figure 2 biosensors-11-00328-f002:**
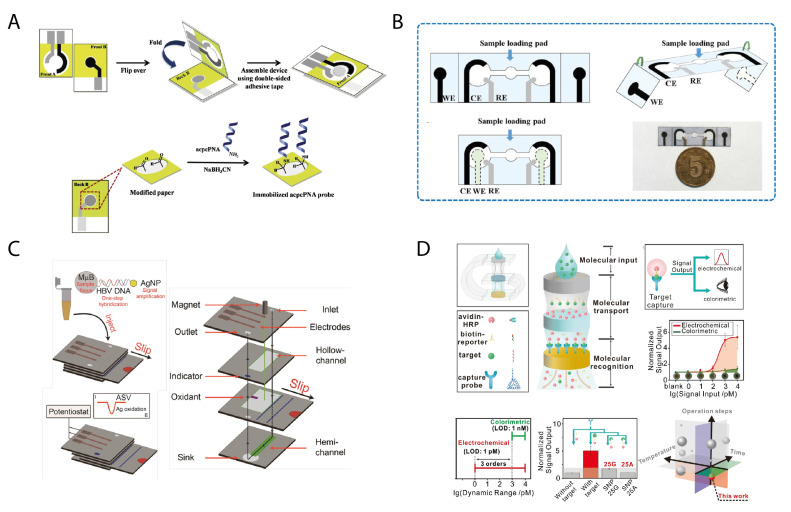
(**A**) Electrochemical impedance-based DNA sensor using pyrrolidinyl peptide nucleic acids for tuberculosis detection. Reprinted with permission from [34], 2018 Elsevier; (**B**) microfluidic origami nano-aptasensor for peanut allergen Ara h1 detection. Reprinted with permission from [35], 2021 Elsevier; (**C**) detection of Hepatitis B Virus DNA with a paper electrochemical sensor. Reprinted with permission from [36], 2015 American Chemical Society; (**D**) molecular threading-dependent mass transport in paper origami for single-step electrochemical DNA sensors. Reprinted with permission from [37], 2018 American Chemical Society.

**Figure 3 biosensors-11-00328-f003:**
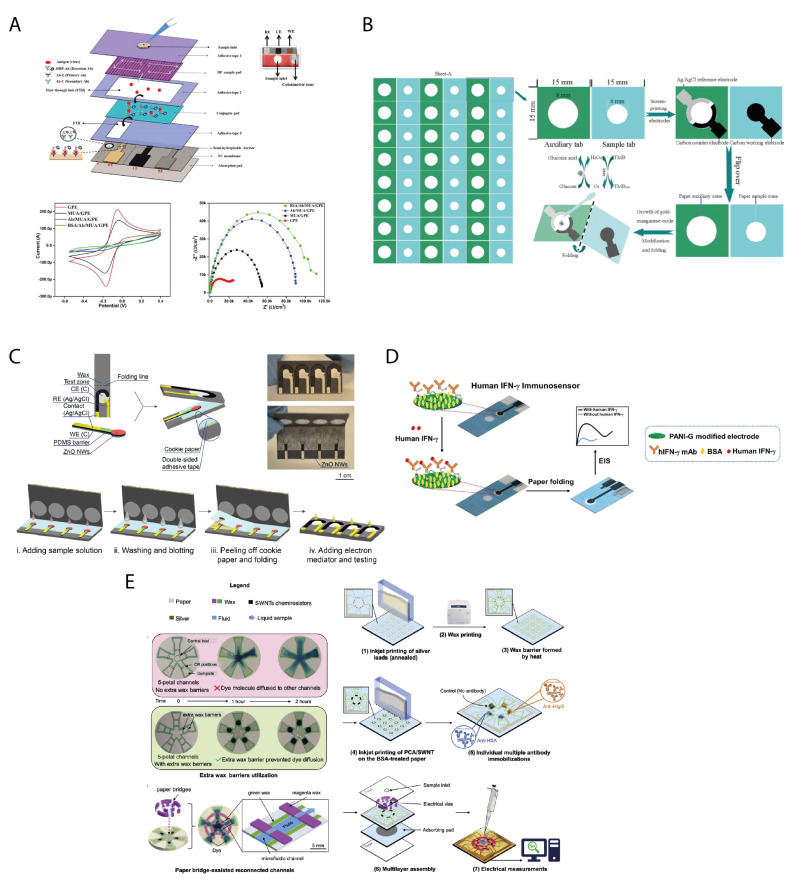
(**A**) Vertical flow-based paper immunosensor for rapid electrochemical and colorimetric detection of influenza virus using a different pore size sample pad. Reprinted with permission from [38], 2018 Elsevier; (**B**) growth of gold-manganese oxide nanostructures on a 3D origami device for glucose-oxidase label based electrochemical immunosensor. Reprinted with permission from [40], 2014 Elsevier; (**C**) a microfluidic paper-based origami nanobiosensor for label-free, ultrasensitive immunoassays. Reprinted with permission from [41], 2016 WILEY-VCH Verlag GmbH & Co; (**D**) label-free paper-based electrochemical impedance immunosensor for human interferon gamma detection. Reprinted with permission from [42], 2018 Elsevier; (**E**) an origami immunosensor for multiplexed analyte detection in body fluids. Reprinted with permission from [44], 2020 Elsevier.

**Figure 4 biosensors-11-00328-f004:**
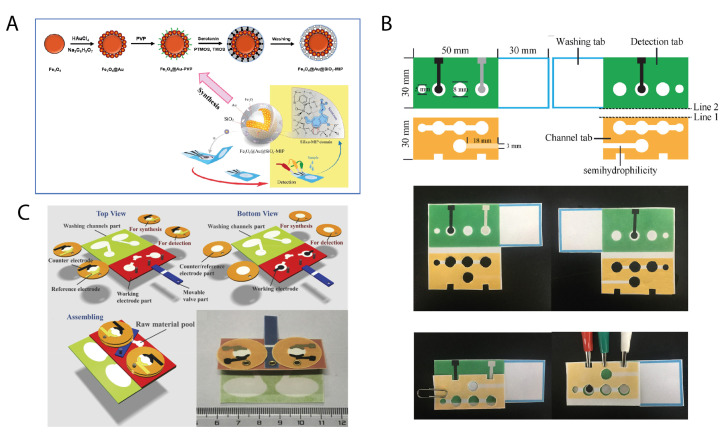
(**A**) Highly sensitive and selective electrochemical paper-based device using a graphite screen-printed electrode modified with molecularly imprinted polymers coated Fe_3_O_4_@Au@SiO_2_ for serotonin determination. Reprinted with permission from [47], 2019 Elsevier; (**B**) ultrasensitive microfluidic paper-based electrochemical/visual biosensor based on spherical-like cerium dioxide catalyst for miR-21 detection. Reprinted with permission from [48], 2019 American Chemical Society; (**C**) the strategy of antibody-free biomarker analysis by in-situ synthesized molecularly imprinted polymers on movable valve paper-based device. Reprinted with permission from [49], 2019 Elsevier.

**Figure 5 biosensors-11-00328-f005:**
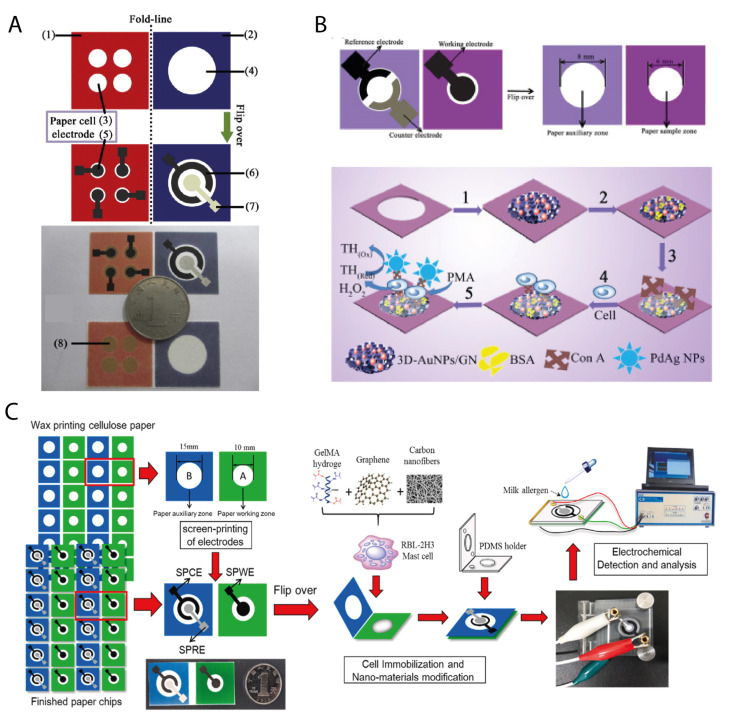
(**A**) Paper-based electrochemical cyto-device for sensitive detection of cancer cells and in situ anticancer drug screening. Reprinted with permission from [50], 2014 Elsevier; (**B**) electrochemical K-562 cells sensor based on origami paper device for point-of-care testing. Reprinted with permission from [51], 2015 Elsevier; (**C**) a novel electrochemical mast cell-based paper biosensor for the rapid detection of milk allergen casein. Reprinted with permission from [52], 2019 Elsevier.

**Figure 6 biosensors-11-00328-f006:**
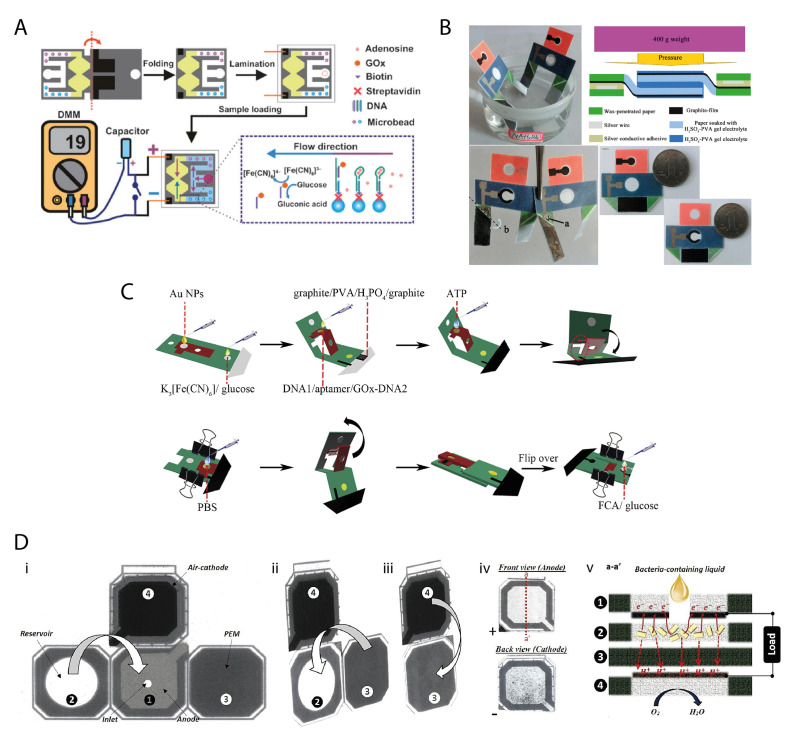
(**A**) Aptamer-based origami paper analytical device for electrochemical detection of adenosine. Reprinted with permission from [53], 2012 WILEY-VCH Verlag GmbH & Co; (**B**) photoelectrochemical lab-on-paper device based on an integrated paper supercapacitor and internal light source. Reprinted with permission from [54], 2013 American Chemical Society; (**C**) a self-powered origami paper analytical device with a pop-up structure for dual-mode electrochemical sensing of ATP assisted by glucose oxidase-triggered reaction. Reprinted with permission from [55], 2019 Elsevier; (**D**) power-on-paper: origami-inspired fabrication of 3D microbial fuel cells. Reprinted with permission from [85], 2017 Elsevier.

**Table 1 biosensors-11-00328-t001:** Summary of the principal features of the origami paper-based electrochemical biosensors.

Type of Origami	Analyte	Sensing (Bio)Component	Electrochemical Technique	Matrix	Linear Range LOD	Ref.
**Origami Paper-Based Electrochemical Enzymatic Biosensors**						
Single folding	Glucose	Glucose oxidase preloaded on paper	Chronoamperometry	Human serum	0–24 mM	[29]
Single folding	Glucose	Glucose oxidase + ferrocenecarboxylic acid	Chronoamperometry	Human blood	1–12 mM 0.05 mM	[30]
Double folding	(a) Glucose (b) Lactate	(a) Glucose oxidase (b) Lactate oxidase	Chronoamperometry	Sweat	(a) 0.08–1.25 mM, 17.05 μM (b) 0.3–20.3 mM, 3.73 μM	[31]
Pop-up	Beta-Hydroxybutyrate	3-hydrozybutyrate dehydrogenase	Chronoamperometry	Human blood	0.1–6.0 mM 0.3 mM	[32]
Multiple folding	Organophosphorus pesticides	Inhibition of Butyrylcholinesterase, with a solid-state butyrylcholine-sensitive membrane	Potentiometry	Standard samples	0.1 and 1.0 nM 0.06 nM	[33]
Multiple folding	(a) Paraoxon (b) 2,4-dichlorophenoxyacetic acid (c) atrazine	Inhibition of: (a) butyrylcholinesterase (b) alkaline phosphatase (c) tyrosinase	Chronoamperometry	River water	(a) 2 ppb (b) 50 ppb (c) 10–100 ppb	[10]
Single folding	Sulfur mustard	Inhibition of choline oxidase	Chronoamperometry	(a) liquid and (b) aerosol sulfur mustard	(a) 1 mM (b) 0.019 g/m^3^	[12]
**Origami paper-based electrochemical DNA sensors**						
Single folding	Mycobacterium tuberculosis ssDNA	Pyrrolidinyl peptide nucleic acid (acpcPNA) + [Fe(CN)_6_]^3−/4−^	Electrochemical impedance spectroscopy (EIS)	PCR-amplified DNA from blood samples	2–200 nM 1.24 nM	[34]
Double folding	Peanut allergen Ara h1	Synthetic aptamer against Ara h1 immobilized on graphene	Differential pulse voltammetry (DPV)	Cookie dough sample	50–1000 ng/mL 21.6 ng/mL	[35]
Multilayer (slipping layer)	ssDNA from hepatitis B virus	Complementary ssDNA #1 immobilized on MBs + Complementary ssDNA #2 labelled with AgNPs	Anodic stripping voltammetry	Standard solutions	85 pM	[36]
Multiple folding	Complementary ssDNA	Complex formed by Biotin-DNA, avidin-HRP, and tetrahedral DNA	Chronoamperometry	Human serum	1–10 pM 1 pM	[37]
**Origami paper-based electrochemical immunosensors**						
Multilayer	Influenza virus H1N1	Horseradish peroxidase tagged antibody- H1N1 -primary antibody complex	Electrochemical impedance spectroscopy	Saliva	0–10,000 PFU/mL 4.7 PFU/mL	[38]
Multilayer	17β-estradiol	Monoclonal antibody immobilized on a MWCNTs/THI/AuNPs coated working electrode	Differential pulse voltammetry	Serum	0.01–100 ng/mL 10 pg/mL	[39]
Single folding	Prostate protein antigen	Carbon nanosperes-glucose oxidase-monoclonal antibody label	Differential pulse voltammetry	Serum	0.005–100 ng/mL 0.0012 ng/mL	[40]
Single folding	HIV p24 antigen	p24 antibody immobilized on zinc oxide nanowires coated working electrode	Electrochemical impedance spectroscopy	Serum	300 fg/mL	[41]
Single folding	Human interferon-gamma	Monoclonal human antibody was immobilized on polyaniline modified graphene electrode	Electrochemical impedance spectroscopy	Serum	5–1000 pg/mL 3.4 pg/mL	[42]
Multiple folding	(a) Human chorionic gonadotropin (b) prostate-specific antigen (c) carcinoembryonic antigen	Reduced graphene oxide/Ag@BSA/secondary antibody as signal label.	Cyclic voltammetry	Serum	(a) 0.002–120 mIU/mL, 0.0007 mIU/mL. (b) 0.001–110 pg/mL, 0.35 pg/mL. (c) 0.001–100 pg/mL, 0.33 pg/mL	[43]
Multiple folding	Human serum albumin and human immunoglobin G	Monoclonal anti-human antibodies immobilized onto pyrene carboxylic acid-modified single-walled carbon nanotubes	Chemiresistor	Urine, saliva, serum and whole blood	1.5 pM	[44]
Multilayer (slipping layer)	Ricin *a* chain	MB/anti-ricin *a* chain antibody/ ricin *a* chain/monoclonal mouse anti-ricin *a* chain antibody/AgNP immunocomposite	Anodic stripping voltammetry	Standard solution	34 pM	[45]
Multilayer (slipping layer)	Trefoil Factor 3	MB/spAb-TFF3-mpAb/2°mpAb/AgNP immunocomplex	Anodic stripping voltammetry	Urine	0.03–7.0 μg/mL	[46]
**Origami paper-based electrochemical MIP sensors**						
Single folding	Serotonin	Fe_3_O_4_@Au@SiO_2_-MIP nanocomposite	Linear sweep voltammetry (LSV)	pharmaceutical and urine samples	0.5–1000 μM 0.08 μM	[47]
Double folding	Glycoproteins	SiO_2_@Au/dsDNA/CeO_2_ nanocomposite	Differential pulse voltammetry (DPV)	Egg white samples	1–10^7^ pg/mL 0.87 pg/mL	[48]
Single folding + rotating elements	Carcino-embryonic antigen (CEA)	GO/chitosan/ glutaraldehyde/ dopamine	Differential pulse voltammetry (DPV)	Human serum	1.0–500.0 ng/mL 0.32 ng/mL	[49]
**Origami paper-based electrochemical cell-based biosensors**						
Single folding	Human acute promyelocytic leukemia cells (HL-60)	Aptamer KH1C12	Differential pulse voltammetry (DPV)	Human serum	5.0 × 10^2^–7.5 × 10^7^ cells/mL 4 cells/10 μL	[50]
Single folding	Human chronic myelogenous leukemia cells (K-562)	Concanavalin A immobilized on IL/3D-AuNPs/GN/composite	Differential pulse voltammetry (DPV)	Standard solutions	1.0 × 10^3^–5.0 × 10^6^ cells/mL 200 cells/mL	[51]
Single folding	Casein allergen	Rat basophilic leukemia (RBL-2H3) mast cells	Differential pulse voltammetry (DPV)	Standard solutions	10^−7^–10^−5^ g/mL 32 ng/mL	[52]
**Self-powered origami paper-based electrochemical biosensors**						
Single folding	Adenosine	Biotin-labeled aptamers immobilized on streptavidin-functionalized MBs	Electrochemical readout with a digital multimeter	Standard solutions	Up tp 250 μM 11.8 μM	[53]
Multiple folding	Adenosine triphosphate	ssDNA immobilized on a chemiluminescence-photoelectrochemical system composed of ABEI−AuNPs, p-iodophenol, and thioglycolic acid-capped CdS NPs	Electrochemical readout with a digital multimeter	Human serum	1–1000 pM 0.2 pM	[54]
Pop-up	Adenosine triphosphate	Aptamer hybridized with GOx-labelled ssDNA and ssDNA immobilized on AuNPs	Electrochemical readout with a digital multimeter and Differential pulse voltammetry (DPV)	Standard solutions	10–5000 nM 3 nM	[55]

AgNPs, silver nanoparticles; MBs, magnetic beads; MWCNTs, multiple wallet carbon nanotubes; THI, thionine; AuNPs, gold nanoparticles; Ag@BSA, bovine serum protein-stabilized silver nanoparticles; spAb, monoclonal mouse anti-human TFF3 solid-phase Ab; mpAb, monoclonal rabbit anti-human TFF3 mobile-phase antibody; 2° mpAb, Biotinylated goat anti-rabbit secondary mobile-phase antibody; dsDNA, double strand DNA; GO, graphene oxide; IL, ionic liquid; GN, graphene; ABEI, N-(aminobutyl)-N-(ethylisoluminol); ssDNA, single strand DNA sequence; CdS NPs, cadmium sulfide nanoparticles; GOx, glucose oxidase.

## Data Availability

The data presented in this study are available on request from the corresponding author.

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
