# Peer review of "Origami Paper-Based Electrochemical (Bio)Sensors: State of the Art and Perspective"

_biosensors, 2021, doi:10.3390/bios11090328_

Round 1

Reviewer 1 Report

The manuscript presents a review of origami electrochemical paper-based biosensor and classify the results in 5 categories: enzymatic, DNA, immunosensors, molecularly imprinted polymer, cell-based and self-powered origami paper-based sensors. Through the manuscript, several fundamental concepts regarding the operation of the sensors were presented compactly and understandably. 

Although the information is well presented, the text would also be presented as a table with the main characteristics listed as receptor, analyte, electrochemical technique, analytical parameters and reproducibility. This format could make the manuscript more readable and easy- to follow for readers.  

The authors could add information about the search engine ( scopus, scholar google, web of science) and their keywords during the bibliographical search. They also can provide information about the limits in time for the search (for example, from 2010 to the present). 

I think a schematic diagram or at least a graphical abstract of origami-based electrochemical sensor would be helpful. Particularly,  a figure with the electrochemical origami-based device in the center and enzyme, DNA, antibody, MIP and cell around it would be illustrative for the topic of the review.

Reviewer 2 Report

General Comment

In this review manuscript the authors report on the last advances on the use of paper-based electrochemical (bio)sensors, focusing on the 3D analytical devices (origami (bio)sensors) built from the special foldability feature of paper. The manuscript is well organized in sections according to the type of (bio)sensor (enzymatic biosensors, DNA biosensors, immunosensors, MIPs, cell-based sensors), and closing the main text with an important section about self-powered paper-based biosensors. Throughout the text, outstanding works in the area are critically described. It is worth emphasizing the expertise of the authors in the specifical area of the proposed review article. Based on these considerations, I suggest the acceptance of the manuscript for publication in the Biosensors journal after minor revisions. Following are some specific comments to further improve the quality of the manuscript.

Specific Comments

  1. Introduction, page 2: please, rephrase the following sentence “…the analytical chemistry could carry out many activities to carry out to achieve the different SDGs,…”;
  2. Section 2, page 3: please, provide more detail on the enzymatic immobilization procedure in this sentence “Then the bond of the enzymatic pad to the working electrode…”, ref. of Liang et al [19];
  3. Section 2, page 3: in the case of the work reported by Ding et al., how was the cell potential generated for the potentiometric measurement? Has a Nernstian behavior been verified? What about selectivity tests?
  4. Section 3, page 7: about the discussion of the work of ref. [45] (from Henry’s group), I would suggest the authors describe the process applied for paper oxidation (sentence "The origami sensor was developed by covalently immobilizing a pyrrolidinyl peptide nucleic acid (acpcPNA) onto partially oxidized cellulose paper");
  5. Section 4, page 10: when discussing the work of Li et al. [60], the authors mention “Also in this case, paper was exploited for the in situ growth of zinc oxide nanowires directly on a carbon working…”; How was the in situ synthesis of zinc oxide nanowires on paper carried out? It's an interesting approach that could be better detailed;
  6. Section 4, page 11, line 492: I think something is missing in the following sentence “…for the simultaneously and selectively detection of different analytes [62] proposed an origami multiplexed….”;
  7. Section 4, page 13: Can the resolution of Fig. 3c be improved?
  8. Section 5, page 14: please, check the following sentence “Amatatongchai et al. [74] applied a simple origami configuration of origami…”;
  9. What is the authors' perception on the problem of biofouling in the use of paper-based electrochemical (bio)sensors? How can this be overcome using this technology?
  10. In the “Conclusions and Perspective” section, I believe it would be interesting if the authors comment on the current and future scenario regarding the commercialization of paper-based electrochemical devices. What obstacles need to be overcome for the wide commercial use of the devices?

Reviewer 3 Report

This paper is a review paper about origami-based biosensors. 

  1.  The first part of the introduction (Line 32-58) is verbose. It is better to summarize the contents with a reference.
  2. Line 104-118. It is better to remove this part or summarize it with references.
  3.  A couple of titles should be revised as follows: A) Origami paper-based enzymatic biosensors -> Origami paper-based electrochemical enzyme biosensors / B) Origami paper-based cell-based sensors -> Origami paper-based electrochemical cell-based biosensors / C) Self-powered paper-based biosensors -> Self-powered paper-based electrochemical biosensors.
  4. Authors should choose the words carefully, which point out each sensor, and use them consistently. Please review them.
